# Towards an Operative Predictive Model for the Songshan Area during the Yangshao Period

**Lijie Yan** [1,2], **Peng Lu** [1,2,*], **Panpan Chen** [1,2], **Maria Danese** [3], **Xiang Li** [4], **Nicola Masini** [3,5], **Xia Wang** [1,2], **Lanbo Guo** [1,2] **and Dong Zhao** [1,2]

1   Institute of Geography, Henan Academy of Sciences, Zhengzhou 450052, China; lijiey1213@sina.com (L.Y.); cppbed@163.com (P.C.); 15600624913@163.com (X.W.); guolanbo_luoyang@163.com (L.G.); zdong1102@Outlook.com (D.Z.)
2   Zhengzhou Base, International Center on Space Technologies for Natural and Cultural Heritage under the Auspices of UNESCO, Zhengzhou 450052, China
3   Institute of Science of Cultural Heritage, National Research Council, C.da Santa Loja, 85050 Tito Scalo (PZ), Italy; maria.danese@cnr.it (M.D.); n.masini@ibam.cnr.it (N.M.)
4   School of Earth and Environmental Sciences, University of Queensland, Brisbane, QLD 4072, Australia; xiang.li11@uqconnect.edu.au
5   Key Laboratory of Digital Earth Science, Institute of Remote Sensing and Digital Earth, Chinese Academy of Sciences, No.9, Dengzhuang South Road, Haidi, Beijing 100094, China
*   Correspondence: bulate_0@163.com

**Abstract:** The literature in the field of archaeological predictive models has grown in the last years, looking for new factors the most effective methods to introduce. However, where predictive models are used for archaeological heritage management, they could benefit from using a more speedy and consequently useful methods including some well-consolidated factors studied in the literature. In this paper, an operative archaeological predictive model is developed, validated and discussed, in order to test its effectiveness. It is applied to Yangshao period (5000–3000 BC) in the Songshan area, where Chinese civilization emerged and developed, and uses 563 known settlement sites. The satisfactory results herein achieved clearly suggest that the model herein proposed can be reliably used to predict the geographical location of unknown settlements.

**Keywords:** GIS; prediction model; archaeology; Songshan; Yangshao period

## 1. Introduction

Over the years, numerous archaeological predictive models have been developed to study the existing relationships between environmental parameters and known archaeological site locations [1,2]. This was done in order to assess the likelihood of finding remaining archaeological sites containing the past human activity [3–6], and also for management and protection reasons. GIS is important for understanding and summarizing spatial relationships, and it offers the potential to exploit this knowledge to structure solution techniques and new location models [7]. GIS predictive models enable archaeologists to test a theory through the use of empirical data, and are generally used to detect archaeological sites by taking into account statistical samples or anthropologic dynamics [8]. The Integrated Conservation of Cultural Landscape Areas [9] recommended the use of predictive modeling through statistical analysis to infer the occurrence of sites on the basis of observed patterns and assumptions about human behavior [10].

In fact, traditionally observed settlement patterns and assumptions related to the relationships between natural and social environmental parameters have been statistically investigated to obtain "settlement rules" that are important to improve the understanding of past human behavior and develop interpretations of the socio-economic structures of past societies [11].

However, over time, different approaches and methodologies have been used to set predictive models, implemented using a heterogeneous set of parameters and statistical analysis tools to tailor the models and validate the outputs. There are two main predictive modelling approaches, generally known as: (i) inductive and (ii) deductive methods [12–15]. Inductive methods are the most common, and are based on the extraction of rules from a well-known dataset or obtained from field surveys. In deductive modeling, the location rules are derived from theoretical knowledge supports, which makes the understanding and interpretation of the results easier [16].

The selection of parameters considered in the model depends on many factors, such as cultures, locations, and historical periods, as well as on the methodological approach adopted. In the specialized literature, there is great variability in the parameters used [17], some of them are site-dependent, i.e., directly linked to the specific archaeological characteristics of the area, while others are site-independent, i.e., linked to environmental or biophysical characteristics as slope, relief, aspect, soil type, elevation, vegetation, distance to water, proximity to food source, etc. The most common applications are generally based on land use, geomorphological parameters such as elevation, slope, and landforms [18], and distances from water bodies [19], disregarding other environmental and social parameters [1,20] such as sun exposure, viewshed analysis, and distances between sites. Additionally, it should be considered that some variables, such as past land use and availability of resources (water, arable land, etc.), are critical to be considered due to the significant differences between current and past conditions.

Other differences are found also in the statistical tools adopted to tailor the models and validate their outputs. The most common methods can vary from simpler methods, such as map algebra functions [21] and use of a weighted composite index [22], to more statistically robust methods, such as Markov's Chain [5], Dempster–Shafer's belief theory [23], logistic regression analysis [24] and multi-fractal approaches [25]. A weighted composite index considers that each environmental variable contributes in a different way to predict the presence or absence of archaeological sites and, therefore, the variables considered are weighted differently on the basis of their impact on the features modeled.

Another critical issue is validation, which has been widely discussed for more than 20 years [26–28] and it is still debated today. Validation is generally performed by comparing the results with independent data sets, as, for example:

- Locations of known archaeological sites; and
- Surveys, in areas classified as having high or moderate probability of storing ancient remains.

To construct an archaeological predictive model is a complex task, because the right method, appropriate parameters, and robust validation criteria must be chosen. All these elements allow us to construct a solid, but time-consuming, support decision system for scholars and actors in the fields of preventive archaeology and archaeological heritage management and protection. Having ascertained the usefulness of these models, sometimes it is necessary to support the work of authorities with the setup of faster and consequently more operative methods that will be, as reliable as possible.

In this paper, an operative method to model settlement location preference was tested for investigation at the regionalization level. The prediction model was established using 563 settlement sites and tested using the locations of additional 55 known sites (not included in the modelling and analysis steps) located around the Songshan area (Figure 1) during the Yangshao period (5000–3000 BC), where Chinese civilization emerged and developed.

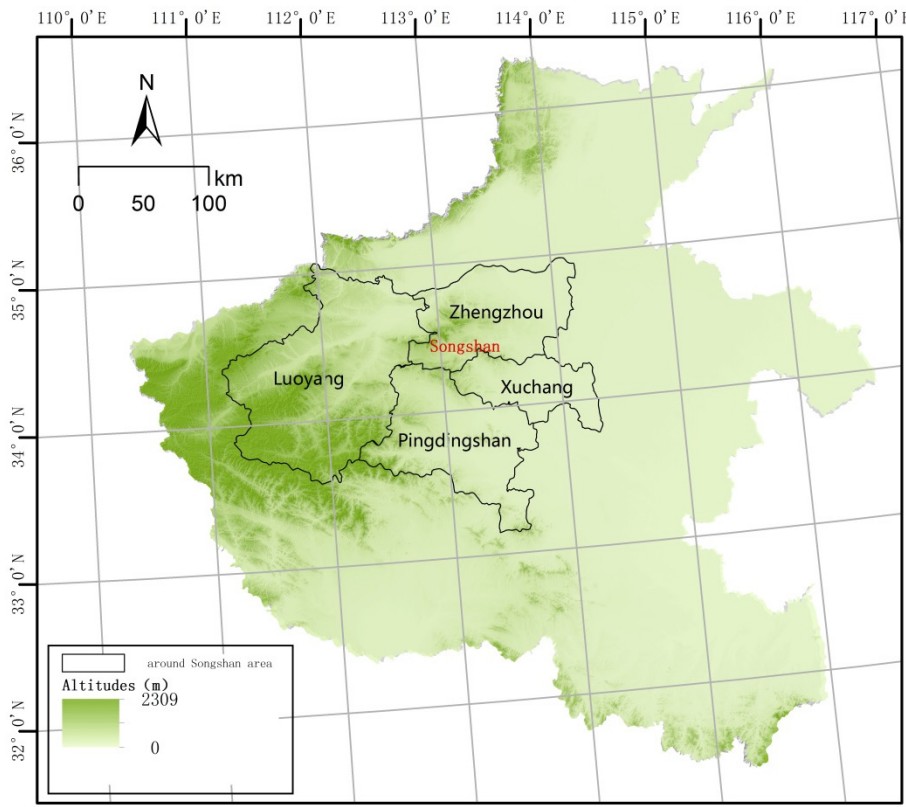

**Figure 1.** The location around the Songshan area.

## 2. The Archaeological Sites of the Yangshao Period

### 2.1. Study Area and Data Acquisition

The region of interest (ROI), located in Songshan, including Zhengzhou, Luoyang, Xuchang, Pingdingshan, is located between 33°6′50″ N–35°3′30″ N, 111°8′20″ E–114°19′20″ E. The length of the EW extension is about 294 km, the length of the NS extension is about 214 km. The total area is 36,000 km$^2$, and is characterized by the presence of mountains in the west, lowlands in the east (with Songshan as the center), erosion and loess hills in the south, and a depression basin in the north [29]. Songshan belongs to the Funiu Mountain system and is one of the Five Mountains of China [30]. Over the millennia, these geomorphological characteristics and the presence of the Yellow River facilitated human frequentation and, consequently, a high concentration of sites of cultural interest. Therefore, this region is very important from an archaeological point of view, as it is considered to be the cradle of Chinese civilization. There are many famous sites, such as Zhijidong [31], Lingjing [32], and other Paleolithic sites; as well as Neolithic sites such as Peiligang [33], Jiahu [34], Tanghu [35], Dianjuntai [36], Shuanghuaishu [37], Wangchenggang [38], Guchengzhai [39], etc. For this reason, it is important to develop a predictive model in this area: (i) to improve the knowledge of the settlement dynamics related to environmental parameters; and to (ii) draw a sensibility map useful for understanding where new discoveries could be made, and consequently for the definition of investigation priorities. In particular, the study and prediction of settlement locations in this area during the Yangshao period plays an important role in improve our knowledge of the origin and development of the Chinese civilization.

### 2.2. Characteristics of Yangshao Period Sites and Choice of Parameters

The Yangshao culture took its name from the village Yangshao (Mianchi County, Henan province) where, in 1921, the first important remains and traces of this culture were discovered. The Yangshao culture, in turn, originated from the Peiligang culture, about 7000–5000 years ago. The cultural evolution from the Peiligang to the Yangshao

periods parallels that of agriculture [40], which was primitive in Peiligang period, but strongly developed in the Yangshao period due to the considerably increased variety of grain crops. In addition to the traditional drought-resistant crop millet, rice was also planted in areas with sufficient water resources. So, the Yangshao culture was mainly an agricultural culture.

Settlement sites in this period were characterized by significant variety in dimensions and building characteristics. Houses in larger settlements had a layout characterized by a trench surrounding them. Outside of the settlements, there were cemeteries and kilns. There were mainly two kinds of houses in the village: round and square in the early period, and square in the later period. The walls of the houses were made of grass and mud. The outer surfaces of the walls were wrapped with grass and then burned to improve water resistance [41].

Yangshao culture was an important Neolithic culture widely distributed across numerous sites in the middle and lower reaches of the Yellow River. According to incomplete statistics, there were nearly one thousand settlement sites in Central China [42].

Settlement distribution in the Yangshao period was mainly conditioned by topography, waterways, and climate [43]. According to the latest research [44], during the Yangshao period, monsoon were weakened and precipitation was reduced, which made people more dependent on the river system. The settlement distribution characteristics of Yangshao period Songshan show that human beings entered a farming society [45–47]. Millet agriculture was primarily found in the hills and mesas, while mixed agriculture was practiced in the plains [48]. The most important parameters were geology, slopes, and water accessibility. Moreover, in the relevant research in this area, the visibility between sites was relatively poor, which cannot prove a close connection between sites [49]. Following from all these considerations, the parameters chosen were the following: elevation, slope, distances from rivers, landforms, soils, and climate types.

### 2.3. Choice of Parameters and Data Acquisition

For the purposes of our investigation, a digital elevation model (DEM) (DEM data is provided by Geospatial Data Cloud site, Computer Network Information Center, Chinese Academy of Sciences (http://www.gscloud.cn, accessed on 29 March 2021) with a resolution of 30 × 30 m) (Figure 2a) was used to extract stream and slope data (Figure 2b).

The current river courses were properly rectified (Figure 2c) as suggested by the specialized literature, which enabled us to account for: (i) the changes to the Yi River in the late Pleistocene [50], (ii) the continuous variation of the Yellow River [51] up to the construction of the river networks of the Yiluo and the lower Yellow River (in the Yiluo Basin).

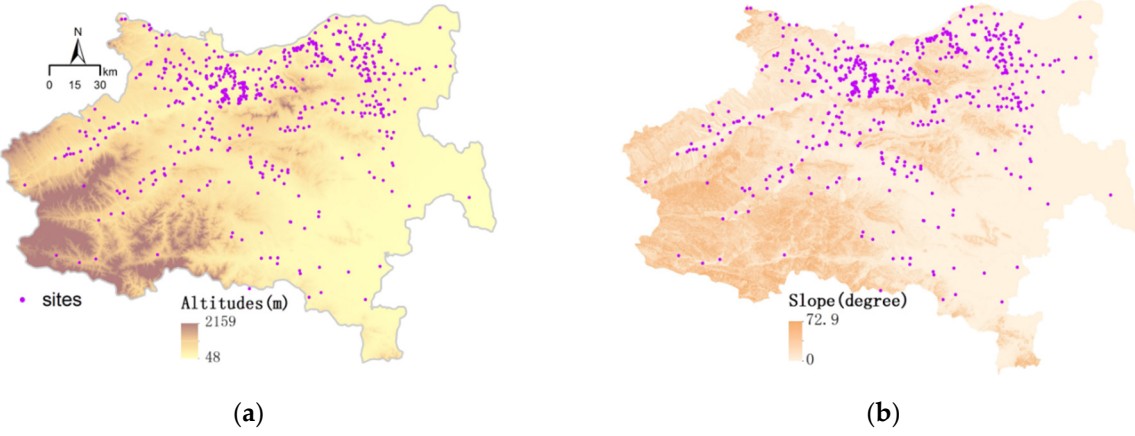

(**a**)                               (**b**)

**Figure 2.** *Cont.*

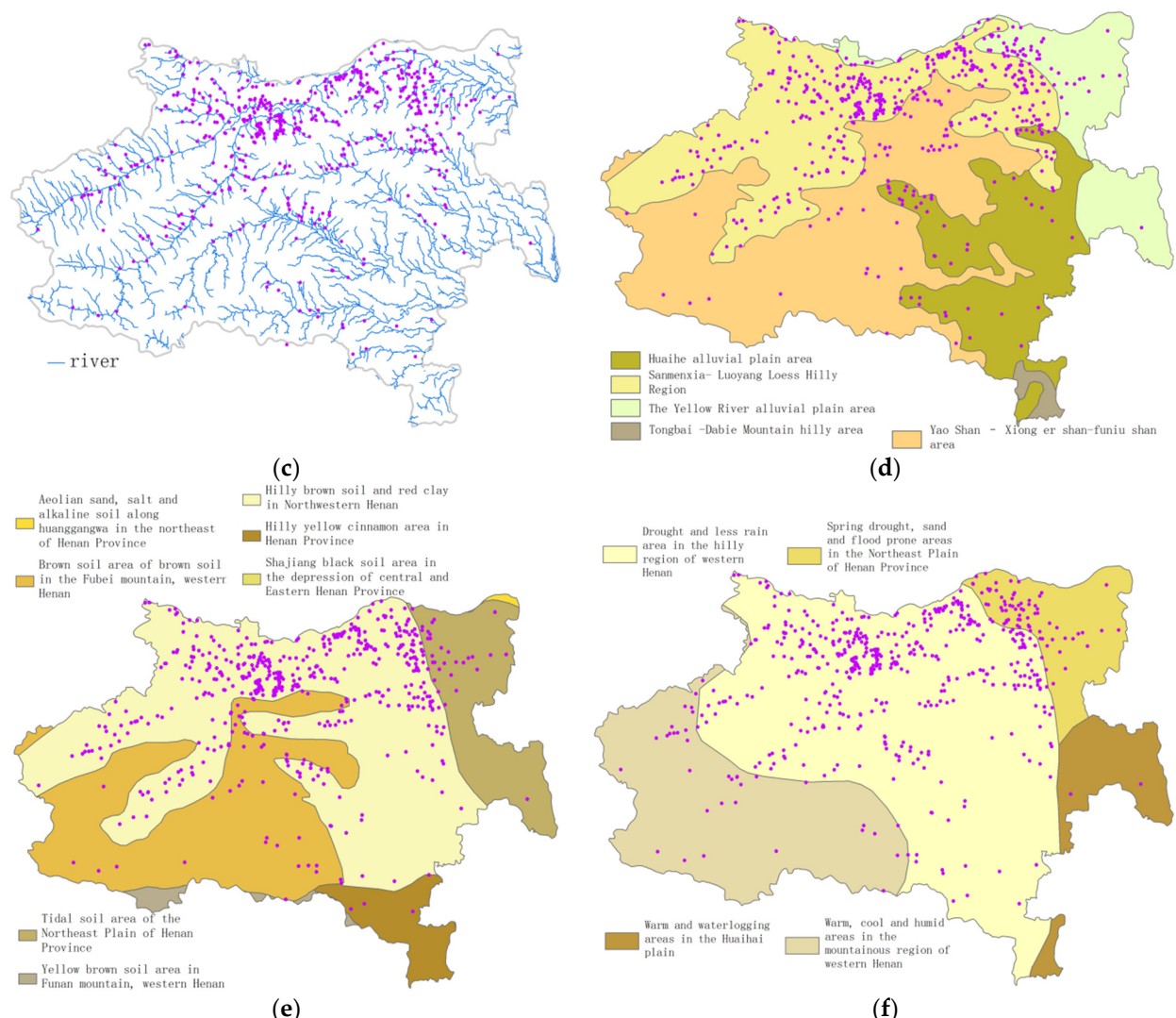

**Figure 2.** Overlay Map of Yangshao period settlements around Songshan with the parameters used in the model: (**a**) altitude; (**b**) slope; (**c**) rivers; (**d**) landforms; (**e**) soil; and (**f**) climate.

Landform, soil, and climate data were obtained from the Atlas of the Agricultural Resources of Henan province [52], at 1:2,500,000 scale (Figure 2d–f).

Data on settlement distribution in the Yangshao period came from the third national survey of cultural relics, the Chinese Cultural Relics Atlas—Henan volume, Henan Province cultural relics and records, and the Yangshao culture site map of Henan province. These data, available in GPS longitude and latitude coordinates, were imported into a GIS environment. The data from other image formats were corrected by the topographic map after registration, and vectorization was carried out with points. In total, 563 settlements were collected and mapped.

## 3. Descriptive Statistics of the Model Parameters

### 3.1. Altitudes

In this area, the lowest elevation was 48 m, and the highest was 2159 m. The study area was divided into eight height elevation ranges for which the site number, site percent for each class, and density of settlements are reported in Table 1.

From Table 1 we can draw the following rules:

1. In areas of below 500 m, the proportion of settlement distribution reached 98.21%.

2. The higher density and distribution of the number of settlements was concentrated in the elevation range between 100–200 m and 200–300 m.
3. In the area of 100–200 m and above, the number and density of settlements decreased with the increase of elevation.
4. At the lowest elevation of 48–100 m, the distribution of the number of settlements and their density was relatively small, indicating that the lowest elevation was not suitable for settlement selection.
5. Yangshao period settlement in the area around Songshan Mountain was mainly distributed in the area with altitude lower than 400 m (see also Figure 2a). It may be that the higher the altitude, the worse the climate, and consequently, those regions were not suitable for human survival.

**Table 1.** Relationship table of settlement distribution and elevation.

| Elevation (m) | Area (km$^2$) | Number (n) | Percent (%) | Density (n/10$^4$ km$^2$) |
|---|---|---|---|---|
| 48–100 | 612.3 | 16 | 2.84 | 261.31 |
| 100–200 | 823.4 | 305 | 54.17 | 3704.15 |
| 200–300 | 470.3 | 150 | 26.64 | 3189.45 |
| 300–400 | 404.25 | 61 | 10.83 | 1508.97 |
| 400–500 | 290.68 | 21 | 3.73 | 722.44 |
| 500–700 | 343.3 | 5 | 0.89 | 145.65 |
| 700–1000 | 322.52 | 4 | 0.71 | 124.02 |
| 1000–2159 | 289.84 | 1 | 0.18 | 34.50 |

*3.2. Slope*

The minimum value of slope in this area was 0; the maximum value was 72.9°. The study area was divided into 12 segments of slope gradient; the number and density of settlement of each slope segment was counted. The results are presented in Table 2 (see also Figure 2b).

**Table 2.** Relationship table of settlement distribution and slope.

| Slope (°) | Area (km$^2$) | Number (n) | Percent (%) | Density (n/10$^4$ km$^2$) |
|---|---|---|---|---|
| 0–1 | 11,194.3 | 181 | 32.15 | 161.69 |
| 1–2 | 4481.24 | 135 | 23.98 | 301.26 |
| 2–3 | 2253.73 | 86 | 15.28 | 381.59 |
| 3–4 | 1740.79 | 51 | 9.06 | 292.97 |
| 4–5 | 1495.27 | 40 | 7.10 | 267.51 |
| 5–6 | 1314.25 | 19 | 3.37 | 144.57 |
| 6–7 | 1179.86 | 13 | 2.31 | 110.18 |
| 7–8 | 1057.86 | 11 | 1.95 | 103.98 |
| 8–9 | 962.24 | 9 | 1.60 | 93.53 |
| 9–10 | 874.51 | 7 | 1.24 | 80.04 |
| 10–15 | 3469.81 | 6 | 1.07 | 17.29 |
| >15 | 5530.31 | 5 | 0.89 | 9.04 |

From Table 2, we can draw the following rules:

1. The site selection mode of prehistoric settlements was in the 0–3° zone, the total number of settlements was 402, accounting for 71.4%.
2. It can be seen from the settlement density that settlement in Yangshao period was mainly concentrated in the 2–3° area, which indicated that the ancients in this period had not completely transferred from the mountains to the plains.
3. The amount and ratio of settlement decreased with the increase of slope, indicating areas with gentle slope were more suitable for settlement. Areas with a greater slope

were less suitable because of the greater cost of settlement construction. Overall, as the slope increased, the density of settlements was constantly reduced (see Table 2).

### 3.3. Distances from Rivers

The early lakes and swamps were mainly distributed in rivers, and the modern river valley is basically the same as the early one. The modern water system pattern essentially reflects the characteristics of the hydrological environment in the early period. The relationship between settlements and distance from the river for intervals of 500 m is reported in Table 3.

**Table 3.** Relationship table of settlement distribution and distance from river.

| Distance from Rivers (m) | Area (km$^2$) | Number (n) | Percent (%) | Density (n/10$^4$ km$^2$) |
|---|---|---|---|---|
| 0–500 | 7376.84 | 276 | 49.02 | 374.14 |
| 500–1000 | 6425.83 | 110 | 19.54 | 171.18 |
| 1000–1500 | 5462.38 | 58 | 10.30 | 106.18 |
| 1500–2000 | 4548.48 | 35 | 6.22 | 76.95 |
| 2000–2500 | 3580.42 | 36 | 6.39 | 100.55 |
| 2500–3000 | 2763.08 | 25 | 4.44 | 90.48 |
| 3000–4000 | 3406.14 | 17 | 3.02 | 49.91 |
| 4000–5000 | 1306.74 | 4 | 0.71 | 30.61 |
| >5000 | 684.36 | 2 | 0.36 | 29.22 |

From the values shown in Table 3, we can deduce the following information:

1.  The areas within 500 m of the river had the largest number of settlements. With an increase in distance from the river system, the number of settlements significantly decreased. This indicates that population had to be close to the river to survive in the Yangshao period. This was because at a low level of productivity, humans had to live near river sources in order to rely on natural runoff.
2.  Most of the settlements were distributed 3 km of the river system (around 96%). Therefore, 3 km seems to be the limit distance within which to live in order to best exploit river resources.

### 3.4. Landforms

The number and distribution of the settlement ratio and the density statistics of each geomorphic area were counted by overlaying the Yangshao period settlements onto the landform map. The results are presented in Table 4.

**Table 4.** Relationship table of settlement distribution and landform types.

| Geomorphic Type | Area (km$^2$) | Number (n) | Percent (%) | Density (n/10$^4$ km$^2$) |
|---|---|---|---|---|
| Sanmenxia–Luoyang Loess Hilly Region | 10,542.06 | 397 | 70.52 | 376.59 |
| Yellow River alluvial plain area | 4594.24 | 58 | 10.3 | 126.25 |
| Huaihe alluvial plain area | 6506.49 | 51 | 9.06 | 78.39 |
| Xiaoshan mountain–Xiongershan mountain–Funiushan mountain area | 13,563.62 | 57 | 10.12 | 42.03 |
| Tongbai–Dabie Mountain hilly area | 347.77 | 0 | 0 | 0 |

From Table 4, we can see that the number and density of settlements were the highest in the Sanmenxia–Luoyang loess hilly area. This showed that in the prehistoric period, the Sanmenxia–Luoyang loess hilly region was the area most suitable for settlement location. In particular, the number of settlements in the Xiaoshan mountain–Xiongershan mountain–Funiushan mountain area was more than that of the Huaihe alluvial plain area, which was equivalent to the Yellow River alluvial plain area, but the density was much lower. There was no settlement in Tongbai–Dabie mountain hilly area, indicating that the mountain and

hilly areas were not suitable for site location due to the complex morphology of the terrain, which was not conducive to human production and life.

### 3.5. Soils

The number, proportion and density of settlements in each soil area determined by overlaying settlements and soil type, shown in Table 5.

**Table 5.** Relationship table of settlement distribution and soil.

| Soil Type | Area (km²) | Number (n) | Percent (%) | Density (n/10⁴ km²) |
|---|---|---|---|---|
| Hilly brown soil and red clay in northwestern Henan | 18,503.80 | 485 | 86.15 | 262.11 |
| Tidal soil area of the northeast plain of Henan Province | 4327.71 | 30 | 5.33 | 69.32 |
| Brown soil area of the north mountain area of western Henan | 10,463.01 | 44 | 7.82 | 42.05 |
| Hilly yellow cinnamon area in Henan Province | 1828.95 | 4 | 0.71 | 21.87 |
| Yellow brown soil area in Funan mountain, western Henan | 366.98 | 0 | 0 | 0 |
| Aeolian sand, salt, and alkaline soil along Huanggangwa in the northeast of Henan province | 62.40 | 0 | 0 | 0 |
| Shajiang black soil area in the depression of central and eastern Henan province | 1.34 | 0 | 0 | 0 |

From Table 5, we can see that the number and density of settlements in Hilly brown soil and red clay in northwestern Henan were far greater than other types. It showed that the ancients settled in the soil area suitable for the development of agriculture in order to stabilize their life in Yangshao period. There were no settlements in the area of yellow brown soil area in Funan Mountain, the western Henan Aeolian sand, the salt and alkaline soil along Huanggangwa in the northeast of Henan province, or the Shajiang black soil area in the depression of central and eastern Henan province.

### 3.6. Climate

The number, proportion and density of settlements of each soil area are shown in Table 6 by overlaying settlements and climate types.

**Table 6.** Relationship table of settlement distribution and climate types.

| Climate | Area (km²) | Number (n) | Percent (%) | Density (n/10⁴ km²) |
|---|---|---|---|---|
| Drought-prone and less rainy area in the hilly region of western Henan | 20,319.16 | 454 | 80.64 | 223.43 |
| Spring drought, sand, and flood-prone areas in the portheast plain of Henan province | 3381.10 | 71 | 12.61 | 209.99 |
| Warm, cool and humid areas in the mountainous region of western Henan | 9409.53 | 36 | 6.39 | 38.26 |
| Warm and waterlogged areas in the Huaihai plain | 2444.41 | 2 | 0.36 | 8.18 |

According to these statistics, the number and density of settlement distribution were the largest in the drought-prone and less rainy area in the hilly region of western Henan, and the values were much larger than those of other climatic types. In the Yangshao period, the drought-prone and the less rainy area in the hilly region of western Henan was more suitable for human habitation.

### 3.7. Summary of the Influencing Factors of Settlement Location and Their Correlation Analysis

In the Yangshao period, site selection was mainly conditioned by the following six environmental parameters: (i) elevation, (ii) slope, (iii) distance from the river system, (iv) geomorphology, (v) soil, and (vi) climate.

The settlement sites were concentrated in the following areas:

- Elevation around 100 to 200 m;
- Slope around 2–3;
- The (horizontal) distance from the river around 0 to 500 m;
- The preferred geomorphic type was the landform area of the Sanmenxia Luoyang loess hilly region;
- The preferred soil was the hilly brown soil and red clay of northwestern Henan; and
- The climate was the drought-prone and less rainy area of the hilly region of western Henan.

Only 26 sites out of 563 sites satisfied these conditions. Therefore, during the Yangshao period, the ancient people used a different site selection mode, based on environmental parameters, in choosing settlement locations.

A correlation analysis was used to eliminate redundant factors, as well as to capture the degree of closeness between the elements of the geographical environment which influenced the location of settlements in prehistoric times. The correlation analysis of geographical environment factors was carried out using SPSS software [53], and the correlation coefficient among the various factors was expressed by Pearson index R. In general, when the absolute value of R was more than 0.7, it was highly correlated, when the absolute value of R was less than 0.7 and greater than 0.4, it was of moderate correlation. When the absolute value of R was greater than 0.1 and less than 0.4, it was of low correlation. When the absolute value of R was less than 0.1, it was unrelated.

Table 7 shows that elevation was positively related to the geomorphology, soil, and climate data. The slope data were positively related to the river system and soil. The landform data were positively related to the elevation, slope, and soil data. The soil data were positively related to the elevation, slope, landform, and climate data, and the climate data were positively correlated with the altitude, elevation, landform, and soil data. The correlation coefficient was always smaller than 0.4, so they all exhibited low level of correlation, suggesting that the six elements of the geographical environment were relatively independent and, therefore, none of them can be excluded from any analysis.

**Table 7.** Correlation coefficients between the geographic environmental factors.

| Elements of Geographical Environment | Altitude | Slope | Distance from Rivers | Landform | Soil | Climate |
|---|---|---|---|---|---|---|
| Altitude | 1 | 0.03 | −0.001 | 0.317 ** | 0.282 ** | 0.386 ** |
| Slope | 0.03 | 1 | 0.128 ** | −0.055 | 0.11 ** | 0.05 |
| Distance away from river | −0.001 | 0.128 ** | 1 | −0.069 | −0.01 | −0.023 |
| Landform | 0.317 ** | −0.055 | −0.069 | 1 | 0.338 ** | 0.178 ** |
| Soil | 0.282 ** | 0.11 ** | −0.010 | 0.338 ** | 1 | 0.213 ** |
| Climate | 0.386 ** | 0.050 | −0.023 | 0.178 ** | 0.213 ** | 1 |

Note: ** indicates significant correlation at 0.01 level.

## 4. The Development of an Operative Prediction Model of Settlement Location in Yangshao Period around Songshan

### 4.1. Quantification of Influence Factors of Settlement Location

In order to eliminate the inconsistency of the dimensions and diverse units of the environmental parameters, it is necessary to quantify the value of the impact factors through data standardization. The quantitative basis was the relationship between the quantity or density of settlement distribution and the geographical environment.

Density of settlement distribution was used as a quantification standard to account for the influence of elevation, slope, landform, soil, and climate on settlement. The number of settlements was used as a quantification standard to account for the influence of river system.

The formula used to quantify the score is expressed in Equation (1):

$$f_i = \frac{v_i}{v_{\max}} \times 100 \tag{1}$$

$f_i$ was a quantified score in the formula. If the geographic element was a river system, $v_i$ represented the number of prehistoric settlements distributed in the $i$ buffer zone of the river system, and $v_{\max}$ was the maximum value of the number of prehistoric settlements distributed in all the buffer zones of the river system.

If the geographical elements were elevation, slope, landform, soil, and climate, $v_i$ was the prehistoric settlement density of the $i$ segment of a geographical element, and $v_{\max}$ was the maximum of the prehistoric settlement density distributed amongst all the subsections of the given geographical element. The maximum and minimum values of the score were 100 and 0, respectively, so that the value of 100 represented the region with the highest preference for prehistoric settlement, whereas the value of 0 represented the least preferred area. The quantitative results are shown in Table 8.

### 4.2. Weights Determination of Influence Factors of Settlement Location

The weight set reflected the relative importance of each environmental parameter which affected the settlement location. Weighting methods can be divided into three categories: subjective, objective, and combined. The subjective empowerment approach is mainly based on a subjective judgment of experts to obtain the index weight, such as in the Delphi method [54] and the analytic hierarchy process. Even if this approach is widely used, its objectivity is poor, being that it depends on the knowledge, experience, and personal preferences of the experts, i.e., on the emphasis that experts subjectively place on each index, with no consideration for the characteristics of the data under investigation. The objective weighting approach has a strong theoretical basis and objectivity, and it uses diverse methods, such as entropy or variation coefficients, to calculate the index weight, exploiting the relationships among the original data. Therefore, the objective weighting method was easily subject to the influence of the data sample, as well as to the specific method adopted to assess the weights from the available data sample, being that diverse methods tend to yield different results.

In this paper, an objective weighting approach was used to calculate the weights of the diverse environmental and geographic factors which influenced settlement distribution in the Yangshao period. In order to make the result more accurate, we adopted two weighting approaches: (i) the variation coefficient, and (ii) the entropy method. Moreover, to mitigate the limitations of single weighting models, the final weight of the factors influencing settlement location were the average of the weights obtained from both the variation coefficient and the entropy method.

### 4.3. Variation Coefficient

The variation coefficient is an objective weighting method to determine weights using evaluation indices. Compared with the subjective weighting method, this method is more scientific, objective, and reliable [55].

The steps to calculate the weights of the factors affecting settlement site selection are as follows.

Firstly, the coefficient of variation, $C_V$, of each influencing factor was calculated. The formula of the coefficient of variation of each influencing factor was as follows (Equation (2)):

$$C_{Vi} = \frac{\sigma_i}{\overline{x_i}} \tag{2}$$

where $C_{Vi}$ is the $i$ coefficient of variation of the $i$th influencing factor, also known as the standard deviation coefficient, $\sigma_i$ is the standard deviation of the $i$th influencing factor, and $\overline{x_i}$ is the average number of the $i$th influencing factor.

Secondly, the weight of each factor was calculated as follows (Equation (3)):

$$w_i = \frac{V_i}{\sum_{1=1}^{n} V_i}$$

(3)

where $w_i$ represents the weight of the *i*th impact factor, while $v_i$ is the same as in Equation (1).

**Table 8.** Scores with different factors and grades.

| Factors | Different Levels | Quantitative Score ($f_i$) |
|---|---|---|
| Elevation (m) | 48–100 | 7 |
| | 100–200 | 100 |
| | 200–300 | 86 |
| | 300–400 | 41 |
| | 400–500 | 20 |
| | 500–700 | 4 |
| | 700–1000 | 3 |
| | 1000–2159 | 1 |
| Slope (°) | 0–1 | 37 |
| | 1–2 | 79 |
| | 2–3 | 100 |
| | 3–4 | 84 |
| | 4–5 | 82 |
| | 5–6 | 38 |
| | 6–7 | 44 |
| | 7–8 | 25 |
| | 8–9 | 22 |
| | 9–10 | 21 |
| | 10–15 | 10 |
| | >15 | 2 |
| Distance from rivers (m) | 0–500 | 100 |
| | 500–1000 | 40 |
| | 1000–1500 | 21 |
| | 1500–2000 | 13 |
| | 2000–2500 | 13 |
| | 2500–3000 | 9 |
| | 3000–4000 | 6 |
| | 4000–5000 | 1 |
| | >5000 | 1 |
| Landform | Sanmenxia–Luoyang loess hilly region | 100 |
| | Yellow River alluvial plain area | 11 |
| | Huaihe alluvial plain area | 0 |
| | Yao Shan–Xiong er shan-funiu shan area | 21 |
| | Tongbai–Dabie mountain hilly area | 34 |
| Soil type | Hilly brown soil and red clay in northwestern Henan | 26 |
| | Tidal soil area of the northeast plain of Henan province | 16 |
| | Brown soil area in the Fubei mountain, western Henan | 100 |
| | Hilly yellow cinnamon area in Henan province | 8 |
| | Yellow brown soil area in Funan mountain, western Henan | 0 |
| | Aeolian sand, salt and alkaline soil along Huanggangwa in the northeast of Henan province | 0 |
| | Shajiang black soil area in the depression of central and eastern Henan province | 0 |
| Climate type | Drought-prone and less rainy area in the hilly region of western Henan | 4 |
| | Spring drought, sand and flood-prone areas in the Northeast plain of Henan province | 94 |
| | Warm, cool and humid areas in the mountainous region of western Henan | 100 |
| | Warm and waterlogged areas in the Huaihai plain | 17 |

*4.4. Entropy Method*

The entropy method determines the weights according to the amount of information contained: the smaller the entropy, the greater the information provided, and, therefore, the greater the weight associated with the index and the role that factor plays in the comprehensive evaluation [56].

The steps to determine the weights of the impact factors of settlement site selection by the entropy method are as follows: Firstly, the original data matrix was constructed (Equation (4)).

$$R = \left(f_{ij}\right)_{m \times n} = \begin{pmatrix} f_{11} & \cdots & f_{1n} \\ \vdots & \ddots & \vdots \\ f_{m1} & \cdots & f_{mn} \end{pmatrix} \tag{4}$$

where, $m$ is the number of settlements in a certain period, $n$ is the number of influence factors, and $f_{ij}$ is the evaluation value of the $i$th settlement under the $j$th influence factor, as defined in Table 8 and Equation (1).

Secondly, the specific gravity $p_{ij}$ of the factor value of the $i$th settlement under the $j$th influence factor was calculated (Equation (5)).

$$p_{ij} = \frac{f_{ij}}{\sum_{1=1}^{m} f_{ij}} \tag{5}$$

Thirdly, the entropy $e_j$ of the $j$th influence factor was calculated (Equation (6)).

$$e_j = -k \sum_{i=1}^{m} p_{ij} \cdot \ln p_{ij} \tag{6}$$

where the coefficient $k$ is defined as in Equation (7):

$$k = \frac{1}{\ln m} \tag{7}$$

Finally, the entropy weight $ew_j$ of the $j$th influence factor was calculated (Equation (8)):

$$ew_j = \frac{\left(1 - e_j\right)}{\sum_{j=1}^{n}\left(1 - e_j\right)} \tag{8}$$

The larger the entropy weight, the more information the influence factor represents, which means that the influence factor had greater influence on settlement site selection.

Using the above two methods, we calculated the weights of the settlement site selection factors in Yangshao period Songshan, finding the results shown in Table 9.

**Table 9.** Weights of factors affecting settlement selection.

| Influencing Factors | Weights Obtained by Entropy Method | Weights Obtained by Variation Coefficient | Final Weight ($W_i$) |
|---|---|---|---|
| altitude | 0.13 | 0.15 | 0.14 |
| slope | 0.17 | 0.19 | 0.18 |
| river | 0.41 | 0.29 | 0.35 |
| soil | 0.11 | 0.14 | 0.1285 |
| landform | 0.11 | 0.13 | 0.1215 |
| climate | 0.06 | 0.1 | 0.08 |

The weight ranking of each influencing factor was exactly the same in both the variation coefficient and entropy methods. The weights of the influencing factors were from greatest to smallest, the river system, slope, elevation, soil, physiognomy, and climate. It can be concluded that in the Yangshao period the order of importance in the settlement

location rules was, from highest to lowest, the river system, slope, elevation, soil, landform, and climate.

### 4.5. Settlement Location Prediction Model Construction

Before constructing the model, the unit of preference classification was determined. Considering the accuracy of DEM data, a grid of 100m × 100m was selected as the unit of preference classification and also as the cell size of the raster analyzed.

The spatial weighted superposition method was used to construct the preferred grade model of Yangshao period settlement site selection. The effect of each factor was superimposed upon different layers, and finally the graded distribution map of settlement site preferences in Yangshao period Songshan was generated. The model (Formula (9)) was:

$$F = \sum_{i=1}^{n} W_i f_i \tag{9}$$

where $F$ was the comprehensive evaluation score of an evaluation unit, $W_i$ was the weight of the first factor, $f_i$ was the score of the second factor corresponding to the evaluation unit as calculated in Table 8, and $n$ was the total number of factors.

## 5. Results and Model Validation

Six geographic and environmental factors were weighted and superimposed to obtain the comprehensive index distribution map of the settlement preferences in Yangshao period Songshan, with values from 0 to 100. By drawing the frequency distribution histogram of the composite index [57], the index values corresponding to the places where the histogram had obviously changed ere used as the boundaries of different grades to classify the preferred degree of settlement location. The criteria were: 80–100 preferred high-grade areas, 52–79 preferred middle-grade areas, 0–51 preferred low-grade areas (Figure 3). The preferred high-grade area was 2666 km², accounting for 7.5% of the total area, and mainly distributed in the Yihe, Luohe, Yiluo, Jialu, Shuangjihe, Yinghe, Ruhe, and Shahe river basins, 500 m away from the river area. The area of preferred secondary districts was 11,650 km², accounting for 32.8% of the total area, and mainly distributed in preferred high-level areas near the region, including Yiyang County, Yichuan County, Luoyang City, Yanshi City, Mengjin County, Xingyang City, Zhengzhou City, Xinmi City, Xinzheng City, Yuzhou City, Jiaxian County and other areas. The preferred low-grade area was 21,220 km², accounting for 59.7% of the total area, and mainly distributed in the western and southern parts of the region, including Luoning County, Luanchuan County, Song County, Ruyang County, Lushan County, Yexian County, Wugang City, Dengfeng City, most of Gongyi City around Songshan Mountain in central China, Zhongmou County, Changge City, Xuchang City, Yanling County and other regions in eastern China.

The preference model was validated using 55 newly discovered Yangshao settlement sites in the third general survey of cultural relics.

There were 23 sites in the high-grade area, with a density of 86.3/104 km², 28 sites in the middle-grade area, with a density of 24/104 km², and only 4 sites in the low-grade area with a density of 1.9/104 km². The results showed that the density of settlements in the preferred high-grade area was much higher than that of the other two areas, indicating the highest probability of finding Yangshao period settlement sites was in the highest-grade area, followed by the preferred middle-grade area, and the preferred low-grade area was the most difficult area in which to find settlement sites.

Overall, we can know that:

1. Yangshao period settled around Songshan Mountain involved different choices for different environments. The settlement sites were concentrated in the areas where the elevation was within 100–200 m, the slope was between 2–3°, the horizontal distance from the river was within 500 m, the geomorphic type was that of the landform of the Sanmenxia–Luoyang loess hilly area, soil type was hilly cinnamon soil and red

clay in northwest Henan, and the climate type was the arid and rainless hilly area in west Henan.

2. The priority of geographic environmental impact factors in settlement selection in the Yangshao period Songshan mountain area was: river system, slope, elevation, soil, landform, and climate.

3. Settlement prediction results showed that the preferred high-grade area was the area with the highest probability of prehistoric settlement, followed by the middle-grade area, and the low-grade area was characterized by the lowest probability of discovering settlement sites. According to this grade, we can predict which areas contain undiscovered settlements to guide field archaeological investigation, determine the scope of field archaeological investigation more accurately, and to actively excavate archaeological sites.

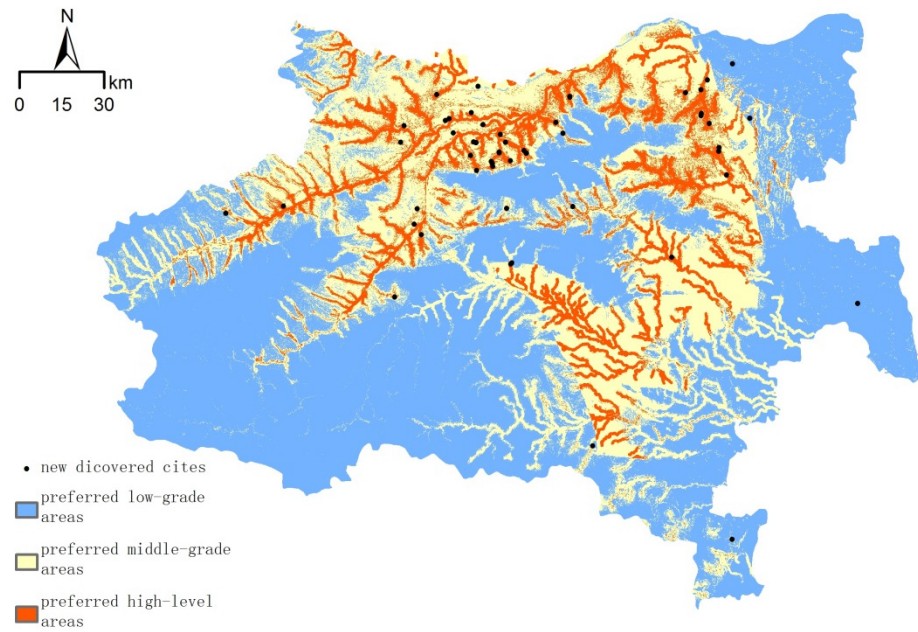

**Figure 3.** Overlay map of the comprehensive zoning of settlement site selection model in Yangshao period Songshan, and new discovered sites.

## 6. Discussion and Conclusions

In this paper, a comprehensive and fast approach for modelling settlement location preferences at a regional level was proposed. The developed method exploits the knowledge related to 563 settlement sites, dated to the Yangshao period of 5000–3000 BC, and located in the Songshan area, where Chinese civilization emerged and developed. Six geographic and environmental factors—elevation, slope, distance from river systems, geomorphology, soil, and climate—were weighted and superimposed to obtain a comprehensive index distribution map of settlement preference.

One of the most important steps in predictive modelling is the calculation of weights which reflect the relative importance of each parameter in the selection process for the identification of settlement locations.

In this paper, the objective weighting approach was used to calculate the weights of the various indices, namely the diverse environmental and geographic factors which influenced the distribution of Yangshao period settlement. In order to make the results more accurate, we adopted two objective weighting approaches: (i) the variation coefficient, and (ii) the entropy method.

The area of investigation was divided into: (i) high-, (ii) middle-, and (iii) low-grade preference zones, and the analysis was carried out exploring the relationship of the 563 settlements with respect to altitude, slope, river, landform, soil and climate. In the model, the

weight of each factor was determined by using the average of the weights obtained using both the variation coefficient and entropy methods.

A settlement location prediction model was obtained using the comprehensive index method and validation was successfully performed using new 55 settlement sites. The results show that the priority order of the factors which affected human settlement was: (I) distance from rivers, (II) slope, (III) altitude, (IV) soil, (V) landform, and (VI) climate. This finding clearly highlighted that the natural environment played a very important role in the choice of settlement location and on its interaction with human activity. In particular, the fact that the most important factors were the distance from rivers and slope are linked to greater resource availability and easier workability of land for agricultural use during the Neolithic period when agricultural techniques were in their early development phase.

As a whole, the outputs from our investigations highlighted that: (i) the location of settlements was not random, but had specific spatial distribution reflecting the regional characteristics of social development; (ii) the combination of Variation Coefficient and Entropy Method made the weighting results more real and reasonable and weakened the influence of abnormal indexes. The satisfactory results herein achieved clearly suggested that the model herein proposed can be reliably used to predict the geographical location of unknown settlements.

Our analysis highlighted that predictive models can fruitfully constitute an important decision-making support system, providing useful information for defining survey priority and facilitating new site discovery, thus saving time and money, especially in large areas. Moreover, predictive models can also contribute to the preservation of archaeological areas and features, serve as witnesses to the human past, and provide useful information for reducing archaeological risks linked to both anthropic and natural risk factors.

To further improve the results from the proposed prediction model, in the future, the authors will explore the possibility of mining the spatial and temporal distribution of prehistoric settlement data, as well as the possibility of using those data as a predictive parameter selection factor. Earth observation technologies such as optical and radar satellite remote sensing and geophysics will also be used [58–61] to detect archaeological proxy indicators.

**Author Contributions:** Data curation, Panpan Chen and Xia Wang; formal analysis, Peng Lu; validation, Maria Danese, Xiang Li, Nicola Masini and Dong Zhao; visualization, Lanbo Guo; writing—review and editing, Lijie Yan All authors have read and agreed to the published version of the manuscript.

**Funding:** This research was funded by the National Natural Science Foundation of China (Grant Nos. 41701014,41971016 and 41671014),the National Social Science Foundation of China (Grant Nos.18CKG003 and 19ZDA227), Science and Technology Project of Henan Province (Grant No. 192102310019), Soft Science Research Project of Henan Province (Grant No. 192400410067), the Study of Environment archaeology in Zhengzhou, the Digital Environment Archaeology Specially-appointed Researcher of Henan, China (Grant No. 210501002), the basic scientific research of Henan (Grant No. 210601027), and the Research on the Roots of Chinese Civilization of Zhengzhou University (XKZDJC202006), and the Science and Technology Think-Tank Project of Henan Academy of Sciences (Grant No. 210701002).

**Institutional Review Board Statement:** Not applicable.

**Informed Consent Statement:** Not applicable.

**Data Availability Statement:** The data presented in this study are available on request from the corresponding author. The data are not publicly available due to privacy reasons.

**Conflicts of Interest:** The authors declare no conflict of interest.

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
