# Peer review of "Towards an Operative Predictive Model for the Songshan Area during the Yangshao Period"

_ijgi, doi:10.3390/ijgi10040217_

Round 1

Reviewer 1 Report

This manuscript presents an expeditious approach to predictive modelling in archaeology. It is an interesting topic characterized by frequent debate, especially in the last twenty years and with the widespread diffusion of geo-spatial analysis tools. The work, given both the relative novelty of the approach and the very interesting case study, can be taken into consideration for publication.

Even if the objectives are described effectively, it seems to me that some aspects are missing, in particular: the data collection methodology used to gather the set used for the model construction it is not discussed; visibility is not taken into consideration as an influence factor on the data; the history of the studies and the different approaches with reference to other cases are not sufficiently discussed in the introduction.

Author Response

Dear Reviewer:

       thank you for  your  constructive comments and suggestions.

Reviewer 2 Report

Please check the full paper to modify the conventions mentioned in the comments

Author Response

Dear  reviewer

    thank  you for your suggestive comments. Please see the attachment.

Reviewer 3 Report

Towards an operative predictive model for the Songshan area during the Yangshao period

The manuscript pursues an Integrated Conservation of Cultural Landscape Areas, to do so, and based on previous authors, they recommend the use of predictive modelling through statistical analysis to infer the occurrence of sites based on observed patterns and assumptions about ‘human behaviour'. 

I have found the work interesting, nonetheless, several issues have arisen while reading it.

  1. In the Introduction, while authors explain the main predictive modelling approaches, among the parameters other authors have used are a food source, a parameter that authors do not use and, for the period under analysis, it is more important than the slope or the distance to rivers. Good geographical hunting zones are not explained, caves, valleys, etc. all these characteristics play a role for settling in a specific place or not.
  2. About the proximity of settlements to rivers, did the authors of the paper think about the possibility of other water bodies, such as lakes, ponds, shallow underground waters or spring rivers at that time? In my opinion, those variables that authors highlight as critical are not well conveyed in the determination of just rivers as a parameter. It can give them an aleatory result. 

According to what I exposed, I think parameters are not the appropriate ones, and the robust validation then fails.

  1. I do not understand what authors are referencing with [30 to 38], do these references provide anything to your research? The exact location of settlements? Validation of previous analytical data? Other methods of analysis?
  2. I do not understand the last paragraph of section 2.1. I understand it comes with the sample the journal provides to authors. This speaks badly about the interests of the authors paid on reviewing the whole manuscript before its sending.
  3. By the beginning of section 2.2., it would be useful for the reader to explain what other authors have found on this period and why their data is (or not) useful for your database. [refs 39-40].
  4. Sentences such as line 185 are not well written or at least expressed. “area of less than 500 meters”? areas placed at 500 meters high??
  5. Sentence in lines 194-195 seems not conclusive to me because the statement is an assertion based on data that it is not irrefutable, precisely because location depends as well on the availability of other resources and accessibility.
  6. The language the authors use by referring to “preferences” of settlement, line 206 for example, could be referred by the most number of settlements, I mean, I am sure Neolithic societies did not only relied on “preference” and issues of territorial delimitations, dominions but were also something that forced placements in one site or another. It was not a matter of “preference”. Societies at that time were not choosing freely the placement. There were intruders and other social factors that affected settlement, and this is not studied at all in the paper, lines283-286. Food (hunting), crops availability, it is not only about water.
  7. Lines 221-226, did the authors analyse systems for retaining rainwater? That was another source at that time.
  8. Did you analyse temperatures at that time? Or just the contemporary ones? Line 398.
  9. It is not clear, or I find it not appropriately explained, how the authors validated the study with the other 55 cases. Line 441 on.
  10. Line 450. There is a need for more information about the history and living conditions at that time, the presence of animals, species, depending on the differences of natural environments according to slopes, water, etc. Were those settlements temporary, according to the availability of food, or fix? Several doubts remain.
  11. The explanation in line 460 makes sense, and I understand what the authors refer to justify the validity of the analysis “the eventual provability of discovering settlement sites”.

Round 2

Reviewer 3 Report

Dear editorial board,
According to the authors' review, I do not see some of my previous comments justified appropriately, and some vague responses do not let the reader understand what specificities, linked to the place and the epoch, are not covered and why. 
Authors provide a vague response to questions: 1. Despite being an agricultural society, the area is not geographically explained in terms of main life sources. 2. Here, the authors' response to my comment should be specifically placed in the paper for readers to understand the context of the resources available in the area of study at that time. 8. I still have a minor concern, but it is a concern when the authors refer to "preferences".  I am sorry, but it is not adequate in some context of the paper. 9. The authors should state in the paper what they respond "there is no literature on the rainwater conservation system, because the productivity level of the ancients(...)". This is linked to comment (2) and the authors' assertions' feasibility. 12. As said in the comment (1), this is not clearly conveyed through the paper. 
Kind regards.

Author Response

Dear  Reviewers:

      Thank  you very much  for your suggestions. The  paper  is attached.  and  a point-by-point response  are as follows:  

Question 1

Reviewer question: In the Introduction, while authors explain the main predictive modelling approaches, among the parameters other authors have used are a food source, a parameter that authors do not use and, for the period under analysis, it is more important than the slope or the distance to rivers. Good geographical hunting zones are not explained, caves, valleys, etc. all these characteristics play a role for settling in a specific place or not.

Author response: Thank you for your question! In Yangshao period, the level of agricultural production made great progress,  people no longer moved by hunting, but settled down(P4, Line 156 - 161 in the revised manuscript)。

Reviewer response to author response: . Despite being an agricultural society, the area is not geographically explained in terms of main life sources

Author response:  thank you very much .During the Yangshao period, millet agriculture was primarily found in the hills and mesas while mixed agriculture was practiced in the plains. Relevant literature [50] has been added. (P4, Line 161 - 163 in the revised manuscript)。

Question 2

Reviewer question: About the proximity of settlements to rivers, did the authors of the paper think about the possibility of other water bodies, such as lakes, ponds, shallow underground waters or spring rivers at that time? In my opinion, those variables that authors highlight as critical are not well conveyed in the determination of just rivers as a parameter. It can give them an aleatory result. According to what I exposed, I think parameters are not the appropriate ones, and the robust validation then fails.

Author response: Thank you very much for the problem. The early lakes and swamps are mainly distributed in rivers, and the early water system has a good inheritance with the modern water system, that is, the modern river valley is basically the same as the early one. Therefore, the modern water system pattern basically reflects the characteristics of hydrological environment at that time

Reviewer response to author response: . Here, the authors' response to my comment should be specifically placed in the paper for readers to understand the context of the resources available in the area of study at that time.

Author response: Thank you for your good suggestion.  The  content  have be specifically placed in the paper. (P7, Line 233 - 236 in the revised manuscript)

Question 8

Reviewer question: The language the authors use by referring to “preferences” of settlement, line 206 for example, could be referred by the most number of settlements, I mean, I am sure Neolithic societies did not only relied on “preference” and issues of territorial delimitations, dominions but were also something that forced placements in one site or another. It was not a matter of “preference”. Societies at that time were not choosing freely the placement. There were intruders and other social factors that affected settlement, and this is not studied at all in the paper, lines283-286. Food (hunting), crops availability, it is not only about water.

.

Author response: Thank you for your problem. In Yangshao period, the Central Plains including the study area formed a solid economic foundation of agricultural farming economy, and all social activities centered on agricultural farming, people no longer moved by hunting, but settled down. This is also closely related to the natural environment. Although there are other social factors affecting the location of settlements, from the point of view of the unity of cultural outlook, the location of settlements has a similar preference.

Reviewer response to author response: . I still have a minor concern, but it is a concern when the authors refer to "preferences".  I am sorry, but it is not adequate in some context of the paper

Author response:  I agree with you.   Site preferences have be revised to Site selection mode. (P7, Line 222; P10, Line 295-296,307; P17, Line474-476 in the revised manuscript)

Question 9

Reviewer question: Lines 221-226, did the authors analyse systems for retaining rainwater? That was another source at that time

Author response: Thank you for your constructive suggestions! We agree with your suggestion. At present, for the Yangshao culture around Songshan, there is no literature on the rainwater conservation system, because the productivity level of the ancients was not high at that time, they basically depended on heaven to get water from the river

Reviewer response to author response: . Despite being an agricultural society, the area is not geographically explained in terms of main life sources

Author response: Thank you very much.  Through literature review, we found that according to the latest research, during the Yangshao period, the monsoon was weakened and the precipitation was reduced, which made people more dependent on the river system. . (P4, Line149-165; in the revised manuscript)

Best regards  

Round 3

Reviewer 3 Report

Thanks for your response.